# Current Understanding of the Involvement of the Insular Cortex in Neuropathic Pain: A Narrative Review

**DOI:** 10.3390/ijms22052648

**Published:** 2021-03-05

**Authors:** Ning Wang, Yu-Han Zhang, Jin-Yan Wang, Fei Luo

**Affiliations:** 1CAS Key Laboratory of Mental Health, Institute of Psychology, Chinese Academy of Sciences, Beijing 100101, China; wangn@psych.ac.cn (N.W.); zhangyh@psych.ac.cn (Y.-H.Z.); luof@psych.ac.cn (F.L.); 2Department of Psychology, University of Chinese Academy of Sciences, Beijing 100049, China

**Keywords:** neuropathic pain, insular cortex, analgesic response, receptor, sex differences

## Abstract

Neuropathic pain is difficult to cure and is often accompanied by emotional and psychological changes. Exploring the mechanisms underlying neuropathic pain will help to identify a better treatment for this condition. The insular cortex is an important information integration center. Numerous imaging studies have documented increased activity of the insular cortex in the presence of neuropathic pain; however, the specific role of this region remains controversial. Early studies suggested that the insular lobe is mainly involved in the processing of the emotional motivation dimension of pain. However, increasing evidence suggests that the role of the insular cortex is more complex and may even be related to the neural plasticity, cognitive evaluation, and psychosocial aspects of neuropathic pain. These effects contribute not only to the development of neuropathic pain, but also to its comorbidity with neuropsychiatric diseases. In this review, we summarize the changes that occur in the insular cortex in the presence of neuropathic pain and analgesia, as well as the molecular mechanisms that may underlie these conditions. We also discuss potential sex-based differences in these processes. Further exploration of the involvement of the insular lobe will contribute to the development of new pharmacotherapy and psychotherapy treatments for neuropathic pain.

## 1. Introduction

Neuropathic pain occurs when the somatosensory nervous system is damaged due to disease or trauma. Its main symptoms include hyperalgesia, allodynia, and spontaneous pain. Hyperalgesia and allodynia refer to increased pain perception in response to stimuli that usually do and do not cause pain, respectively [1]. Neuropathic pain is difficult to cure, and the long-term suffering that it causes can affect an individual’s quality of life and psychological state. For this reason, it is often comorbid with mood disorders, such as depression and anxiety [2]. Nonsteroidal anti-inflammatory drugs (NSAIDs) are effective for nociceptive pain, but have little effect on neuropathic pain [3].

Mu-opioid receptor (MOR) agonists, such as morphine, are the gold standard of analgesia for various types of pain, such as cancer and severe acute pain [4]. However, for neuropathic pain, the effect is not as obvious [5]. Some researchers believe that the delta-opioid receptor (DOR) may be an important target for the treatment of neuropathic pain [6,7]. Repeated administrations of SCN80, a DOR agonist, could improve some symptoms of neuropathic pain in rats with chronic constriction injury (CCI) [8]. The exploration of analgesic drugs indicates that there are many difficulties in the treatment of neuropathic pain. In many cases, antidepressants and anticonvulsants are used to relieve the symptoms of neuropathic pain [9].

Neurosurgical inventions are also options for the treatment of neuropathic pain [10], such as lesion of the dorsal root entry zone (DREZ) [11] and motor cortex neuromodulation [12]. Responses to neuropathic pain treatment differ among individuals, and sex-based differences in analgesia responses have been reported [13]. Exploration of the mechanisms underlying neuropathic pain from a broad range of perspectives may aid in the identification of better ways to relieve this condition.

As the insular cortex is relatively hidden, the embedding of a cannula or an electrode in the insular lobe for laboratory animal research is difficult. Thus, the insula is more mysterious to neuroscientists than other cortical regions. The rapid development of imaging technology has enabled researchers to explore the functions of the insula. Anatomic studies have confirmed that the insula is connected with many brain structures, such as the somatosensory cortex, limbic system, and frontal lobe [14]. Imaging studies suggest that the insular functions are diverse and include those related to interoception and emotion, reward and motivation, cognition, and decision-making [15].

Some researchers have suggested that the insula is an important information integration center [16] or an area of cross-modal integration [17]. The insular cortex has also been found to participate in the processing of empathy and awareness, and it may tell us “how we feel now” and even “who we are” [18]. In addition, the insula participates in the processing of pain, a complex multidimensional experience [19]. Studies have found that the insula is involved in the processing of autonomic responses to noxious stimuli [20] and in the affective–motivational component of pain [21]; however, increasing evidence indicates that the involvement of the insula in pain processing is more complex [22,23,24].

The role of the insula in neuropathic pain has been of interest for decades. The historical progress is shown in Figure 1. In 1956, a study reported a patient with neuropathic pain who had some small soft fused lesions in the insular cortex and parietal operculum [25,26]. Although this result was from an autopsy, it still suggests the relationship between the insular lobe and neuropathic pain. The development of imaging technology in the 1990s promoted progress in the research of the insular cortex. In 1995, Hsieh et al. found increased regional cerebral blood flow (rCBF) in the bilateral anterior insula and other brain regions in patients with painful mononeuropathy using positron emission tomography (PET) [27].

In 1999, Treede et al. proposed that the insular cortex may be involved in the affective–motivational dimensions of pain [19]. However, this may not be enough to explain the special role of the insula in neuropathic pain. The processing of synesthesia is an important feature of the insula. A 2006 study reported the synesthesia of neuropathic pain and odor and the associated insular activation using functional magnetic resonance imaging (fMRI) [28]. Since 2007, people have noticed the relationship between analgesia and the insula [29], as well as the molecules related to the insula and neuropathic pain [30]. In the past decade, neural network research related to the insular lobe has also provided a great deal of evidence revealing the pathogenesis of neuropathic pain [31]. In recent years, researchers have found that empathy may affect neuropathic pain and that the insular lobe is also involved [32].

**Figure 1 ijms-22-02648-f001:**
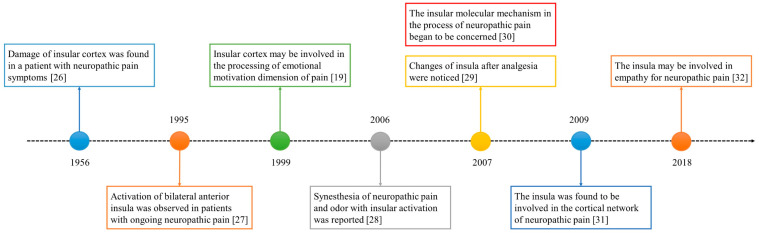
The development timeline of the insular lobe and neuropathic pain.

Although many studies have found that the insular cortex is essential for pain processing, it is still difficult to explain how it participates in pain. In this review, we introduce the structural and functional changes of the insular lobe in the neuropathic pain state and the related molecular mechanisms. There are other brain regions involved in pain, such as the thalamus, primary somatosensory cortex (SI), secondary somatosensory cortex (SII), anterior cingulate cortex (ACC), and frontal cortex [20,33]. Each brain region plays an important and unique role in the occurrence and development of neuropathic pain. However, to clearly describe the insular cortex, we ignore the changes in other brain areas during neuropathic pain.

## 2. Structural and Functional Changes Occurring in the Insular Cortex in the Presence of Neuropathic Pain

### 2.1. Structural Changes

Imaging studies have shown that neuropathic pain can affect the volume or thickness of the insular cortex. Gustin et al. found that the anterior insular gray matter volume (GMV) was reduced, whereas the posterior insular GMV increased in patients with trigeminal neuropathic pain (TNP) [34]. A meta-analysis confirmed that the GMV in the bilateral anterior insula decreased and that in the right posterior insula increased in patients with neuropathic pain [35]. These findings suggest that neuropathic pain can cause distinct anatomical changes in the insular cortex, and these changes may vary in different subregions.

Another study revealed thinning of the cortex of the right dorsal posterior insula and ventral anterior insula in patients with right-side trigeminal neuralgia (TN) [36], suggesting that neuropathic pain affects mainly the thickness of the ipsilateral insular cortex. Such lateralization has also been found in imaging studies of patients with TN [37,38]. However, a meta-analysis revealed that the bilateral insular GMV decreased in patients with TN or TNP [39], indicating that the effect of neuropathic pain on the insular GMV is not limited to the ipsilateral side.

Other types of neuropathic pain can also change the structure of the insula. The GMV of the bilateral anterior insular cortex in patients with spinal cord injury (SCI) was less than that in healthy controls [40]. A meta-analysis of 12 articles reporting on 195 patients showed that SCI can lead to a reduction in GMV in many brain regions, including the insular cortex [41]. Another meta-analysis of 12 GMV datasets representing 190 patients with SCI revealed GMV atrophy in the left insula [42]. In addition, cortical thinning of the insula was observed in patients with ankylosing spondylitis-induced lower back pain [43]. A study using magnetic resonance spectroscopy showed that the *N*-acetylaspartate (NAA) concentration in the insular lobe was significantly decreased in patients with severe traumatic brain injury [44]. The brain NAA concentrations can be used to predict neuron density and activity.

In the above studies, other brain regions also showed reduced GMV, such as the thalamus, SI, putamen, nucleus accumbens [34], right superior frontal gyrus, and left postcentral gyrus [35]. Increased GMV in the right medial frontal gyrus was also observed in patients with neuropathic pain [35]. In patients with SCI, decreased metabolism and GMV were found in the dorsolateral prefrontal cortex [40]. Therefore, the structural changes of the insula during neuropathic pain may be part of the plasticity changes of multiple brain regions. It is still of great value to explore the unique role of the insula in the process of neuropathic pain.

### 2.2. Functional Changes

Numerous imaging studies have demonstrated that the bilateral insula can be activated during pain processing [45,46,47]. The activity of specific brain regions is measured by the changes in certain imaging signals, such as blood oxygenation level-dependent (BOLD) signals in the fMRI or rCBF signals in PET. Not surprisingly, activation of the insular lobe or a part thereof is associated with neuropathic pain. A meta-analysis revealed increased activation of the right caudal anterior insula in the presence of neuropathic pain (fibromyalgia) compared to that observed in the presence of experimental pain [48]. Whether fibromyalgia belongs to neuropathic pain is controversial. In previous studies, fibromyalgia was classified as neuropathic pain, but it has been excluded from neuropathic pain in recent years. Some researchers found that patients with fibromyalgia had pathophysiological changes similar to neuropathic pain [49].

Increased insular activity has also been found in patients with typical neuropathic pain. A recent study showed increased BOLD signals and increased neuronal activities in the bilateral insular cortices immediately after sciatic spared nerve injury (i.e., SNI) in rats using BOLD fMRI and electrode implantation, respectively [50]. In this study, increased activity in the ipsilateral anterior insular cortex was also observed on days 1 and 8 after SNI using manganese-enhanced magnetic resonance imaging (MEMRI) [50]. These results suggest that neuropathic pain has a greater effect on the anterior insula compared to on the posterior insula.

Mechanical and cold allodynia, as the main symptoms of neuropathic pain, can also induce the activation of the insular cortex. In healthy male subjects with peripheral sensitization induced by the subcutaneous injection of ciguatoxins, cold pain hypersensitivity activated the bilateral medial insula [51]. In patients with peripheral nerve injury, brush-evoked allodynia induced the activation of the ipsilateral insular cortex, as observed by PET [52]. An fMRI study also revealed anterior insular cortex activation upon brush-evoked mechanical allodynia in patients with multiple types of neuropathic pain [53]. Thermal hyperalgesia also causes insular activation in certain patients with neuropathic pain. For example, noxious heat stimuli induced more anterior insula activation in patients with diabetic neuropathic pain compared to the controls [54].

Similarly, the increase in insular activity in neuropathic pain is not isolated but is rather accompanied by changes in other pain-related brain regions, including the SI, SII, ACC, and frontal areas [50,51,52]. These brain regions, together with the insula, may form brain networks for pain processing. In these brain networks, the role of the insula and how it works are worth exploring.

### 2.3. Brain Network Involvement

Neuropathic pain studies have revealed increased activities not only in the insula, but also in other related brain regions. The processing of neuropathic pain may involve several brain networks. An fMRI study was conducted to explore the default mode network (DMN) in patients with diabetic neuropathic pain [31]. The DMN refers to the brain network pattern that is activated in the resting state but inhibited during task execution. It may play a role in self-monitoring or self-awareness [55].Using spatial independent component analysis, researchers demonstrated that the bilateral insula is connected to a brain network that is positively associated with spontaneous pain [31]. Spontaneous pain can occur without external stimulation. It is also a symptom of neuropathic pain. Neuropathic pain may be closely related to the abnormality of the DMN, and the insular lobe is involved in this.

The insula may affect the DMN through the interaction between the salience network (SN) and the DMN. Painful stimuli can activate multiple brain regions, including the anterior cingulate cortex and the anterior insular lobe [33,56,57]. Both regions are important nodes of the SN [58], which is a brain network between the DMN in the resting state and the central executive network in the task state. The SN is very important for cognitive control and can be activated by novel sensory stimuli, including pain [59]. A study using resting-state fMRI found that, in multiple sclerosis patients with neuropathic pain, the cross-network communication between SN and other pathways was disrupted [60].

Abnormal functional connections between the thalamus and insula are often observed in the context of neuropathic pain. A study proved decreased gamma-aminobutyric acid (GABA) concentration in the thalamus of patients with neuropathic pain, which was negatively correlated with the strength of the functional connection between the thalamus and insula [61]. Resting-state magnetencephalography revealed spectral changes in the thalamus and posterior insula, with decreased alpha peak power in the DMN, in multiple sclerosis patients with mixed neuropathic pain [62]. Moreover, reduced white matter connectivity between the thalamus and the insula was observed in patients with small-fiber neuropathy [63].

### 2.4. Changes during Analgesia

The activity of the insular cortex can be regulated in the treatment of neuropathic pain [64,65]. In addition, hyperalgesia-induced insular activation, exhibited as prominent BOLD signals, can also be modulated by analgesic and antihyperalgesic effects [29,66]. The thinning insular cortex of patients with TN can be restored to normal by effective treatment [67]. Spinal cord or cortical stimulation can change the activity of some brain regions, including the insula [68,69]. Spinal cord stimulation (SCS) can relieve neuropathic pain and can increase the activation of the contralateral posterior insula [70]. Desipramine, a tricyclic antidepressant, can reduce mechanical allodynia in rats with neuropathic pain induced by selective spinal nerve ligation [71]. Compared to a control group, chronic administration of desipramine can increase the activity of several brain regions, including the insular cortex [71].

[^18^F]-fluorodeoxyglucose PET ([^18^F] FDG-PET) can be used to observe the changes in metabolism and can reflect the activities of brain regions. Using [^18^F] FDG-PET, a study found that transcranial direct current stimulation (tDCS) can relieve the neuropathic pain caused by traumatic spinal cord injury, can increase the metabolism in the insula and the ACC, and can decrease the metabolism in the dorsolateral prefrontal cortex [72]. In addition, motor cortex stimulation (MCS) can be used for the treatment of neuropathic pain [73]. Studies have proven that the rCBF in some brain regions, including the insula and the cingulate gyrus, increase during the “on” status of the MCS, compared to the “off” status [74].

The frontoparietal network, including the frontal and parietal areas, may regulate the activity of the insular lobe in the process of analgesia. The awareness of pain requires not only the processing of the sensorimotor area and limbic system, but also the support of the frontoparietal network, which is crucial for consciousness [75]. The anterior insular cortex is a key node of the SN and is closely connected to the frontoparietal network. Enhanced connectivity between the SN and the frontoparietal network was observed after high-frequency SCS in patients with failed back surgery syndrome [76]. These results support that analgesia can affect the insular activity, and the frontoparietal network may play an important role in it.

To sum up, some typical insular changes in neuropathic pain are summarized in Table 1. GMV reduction and cortical thinning occurring with neuropathic pain, as well as the increased activity caused by allodynia or hyperalgesia, are found mainly in the anterior insula. The GMV in the posterior insular lobe increases in neuropathic pain, suggesting that the role of posterior insular lobe and anterior insular lobe in neuropathic pain may be different.

## 3. Molecular Mechanisms Underlying Neuropathic Pain in the Insular Cortex

Exploring the molecular mechanism of the insula is helpful to reveal how it participates in the processing of neuropathic pain and analgesia. The insular cortex has a variety of receptors, such as opioid, cannabinoid, glutamate, and dopamine receptors. Therefore, the underlying insular mechanisms of neuropathic pain are closely related to neurotransmitters. These receptor molecules are potential analgesic targets. Any change to the binding ability of receptor molecules to ligands will affect their analgesic effect. Neuropathic pain is closely related to central sensitization. Whether the learning- and memory-related molecules in the insular lobe participate in the neuroplasticity and brain reorganization occurring in response to neuropathic pain has become an important research topic in recent years.

### 3.1. Opioid Receptors

Opioid receptors are widely distributed in the spinal cord and cortex, including the insula. The effect of opioid receptors in the insula on analgesia has been examined. A recent study found that naloxone (a nonselective opioid receptor antagonist) and CTOP (a selective MOR antagonist) treatment can attenuate the analgesic effect of NSAIDs injected into the anterior insular cortex on inflammatory pain induced by the plantar injection of formalin in rats [77]. Opioid receptors in the insula may also participate in the analgesic effect of repetitive transcranial magnetic stimulation (rTMS) of the primary motor/sensory region, which can stimulate the release of endogenous opioids [78]. The rTMS at primary motor cortex can relieve neuropathic pain [79]. From the above evidence, opioid receptors in the insular lobe may also participate in this process.

However, the opioid receptor binding ability in the insula may be weakened in certain neuropathic pain states. A PET study conducted with [^11^C] diprenorphine revealed reduced opioid binding in the contralateral insula in patients with post-stroke pain, but not in those with peripheral neuropathic pain [30]. Animal studies yielded similar results. PET demonstrated reduced opioid receptor availability in the insula, and immunohistochemistry revealed reduced insular MOR expression 3 months after SNI in rats [80]. These results suggest that the nervous system injury alters the endogenous opioid system and impairs the MOR binding ability in the insula, which may be one of the reasons why morphine is ineffective in the treatment of neuropathic pain.

Opioid receptors can be divided into several subtypes. In neuropathic pain, the binding rate of the MOR is reduced, but other receptors are not affected. The DOR may be a potential analgesia target for the treatment of neuropathic pain. However, the relationship between the DOR and the insula is still unclear. The kappa-opioid receptor (KOR) in the insula is a potential analgesic target [81]. Therefore, different subtypes of opioid receptors should be considered when exploring the therapeutic effect of insular opioid analgesics on neuropathic pain.

### 3.2. Cannabinoid Receptors

Cannabinoid (CB_1_) receptors are distributed widely in the brain and participate in the regulation of memory, cognition, and other functions, such as analgesia. Similar to the opioid antagonist, microinjection of the CB_1_ antagonist AM-251 into the insular cortex can also inhibit the analgesic effect of NSAIDs in rats with inflammatory pain [77]. The injection of salvinorin A, an agonist of the KOR and CB_1_ receptor, into the insular cortex of rats with sciatic nerve ligature-induced neuropathic pain has a significant analgesic effect, which can be blocked by both KOR and CB_1_ antagonists [81].

Fatty acid amide hydrolase (FAAH) is an enzyme that can hydrolyze the endogenous cannabinoid anandamide. The administration of FAAH can block the analgesic effect of anandamide [82]. URB597 is an inhibitor of FAAH. A study found that the injection of URB597 into the insular cortex of rats with nerve injury-induced neuropathic pain can produce an analgesic effect and can reduce the excitability of insular neurons [83]. These effects are related to the enhancement of endogenous cannabinoid responses.

These findings confirm that the activation of CB_1_ receptors (endogenous cannabinoid or microinjection of cannabinoid receptor agonist) in the insular lobe can produce an analgesic effect in neuropathic pain. Therefore, cannabinoid receptors in the insular lobe may be a potential target for the treatment of neuropathic pain. They may affect pain-related memory and cognition rather than pain sensation.

### 3.3. Dopaminergic Receptors

Dopaminergic receptors, including D_1_ and D_2_, are also found in the insular cortex. Administration of the D_1_ receptor agonist and the D_2_ receptor antagonist in the rostral agranular insular cortex (RAIC) relieved neuropathic pain in rats [84]. The injection of the dopamine reuptake inhibitor GBR-12935 into the RAICs of rats had an antinociceptive effect and decreased c-Fos expression in the spinal dorsal horn [85]. These findings suggest that dopaminergic receptors in the insula are involved in pain relief and that the mesolimbic system plays a regulatory role in the insula.

Neuropathic pain is very common in patients with Parkinson’s disease (PD) [86,87,88]. Using PET, a study proved that pain-induced activation (measured by the rCBF) in the right insula of PD patients was higher than that of a control group [89]. Levodopa, a therapeutic drug for PD, reduces the pain-related insular activation in PD patients. Another PET study showed that levodopa increased the pain threshold and reduced the activation in the right insula induced by cold-pressure pain in patients with PD [90]. These results suggest that neuropathic pain in patients with PD is associated with an impaired dopaminergic system, which affects not only the striatum, but also multiple brain regions, including the insula.

### 3.4. Glutamate Receptors

Nerve injury-induced neural plasticity in the insular cortex may contribute to neuropathic pain, and the long-term potentiation of glutamatergic transmission may also play an important role in this process. In mice with a peripheral nerve injury, an increase in synaptic NMDAR (*N*-methyl-d-aspartate receptor) in the insular cortex was found, and a microinjection of the NMDAR antagonist into the insula was shown to relieve the allodynia-like behaviors [91]. In another study, excitatory synaptic transmission, which is mediated by the AMPAR (aminomethyl phosphonic acid receptor), was enhanced in the insular cortex after peripheral nerve ligation in mice [92]. These findings suggest that ionic and metabotropic glutamate receptors are involved in central sensitization in the insular lobe in the presence of neuropathic pain.

### 3.5. Phospho-Extracellular Signal-Regulated Kinase

Phospho-extracellular signal-regulated kinase (pERK) is an important signaling molecule involved in neuroplasticity and is often used to characterize the activation of brain regions. In rats with chronic constriction injury of the infraorbital nerve, which is a model of trigeminal neuropathic pain, moving and stroking of the infraorbital skin were shown to significantly increase pERK-1/2 expression in the insular cortex [93]. This finding suggests that ERK phosphorylation in the insula is involved in the central sensitization of neuropathic pain.

Increased expression of pERK in the insular cortex was also observed in rats with neuropathic pain induced by SNI of the sciatic nerve [94]. In a mouse model involving corneal alkali burn, increased expression of ERK in the insular cortex was observed, and its blockade reduced spontaneous corneal pain [95]. The administration of the ERK inhibitor in the insular lobe relieved nociceptive behaviors and negative emotions in a rat model of orofacial neuropathic pain [96].

### 3.6. Mechanistic Target of Rapamycin 

Rapamycin is a specific inhibitor of the mechanistic target of rapamycin (mTOR). Signaling by mTOR can influence synaptic plasticity. In recent years, significant progress has been made in the use of rapamycin to treat neuropathic pain. Animal experiments have shown that the microinjection of rapamycin into the insular cortex can reduce the neuronal activity and can inhibit the synapse plasticity induced by neuropathic surgery [97,98,99]. Immunohistochemical analysis revealed that rapamycin reduced insular cortex activity in a mouse model of persistent postsurgical pain [97]. Other mTOR inhibitors, such as Torin1 and XL388, can also reduce insular activity and can relieve neuropathic pain [98,99].

The mTOR signaling pathway can regulate the autophagy [100]. Studies found that autophagy has a protective effect on the nervous system after nerve injury [101,102]. Besides, the dysregulation of autophagy may promote the occurrence of neuropathic pain [103,104]. Ambar1 (activating molecule in Beclin1-regulated autophagy) is an autophagy regulatory protein [105]. The heterozygous Ambra1 transgenic mice (i.e., Ambra1 mice) have autophagy deficiency. A study used CCI of the sciatic nerve to establish neuropathic pain in Ambra1 mice and demonstrated that autophagy deficiency in these mice can lead to metabolic dysfunction and aggravation of neuropathic pain [106]. Caloric restriction, used to induce autophagy, can fight against the development of neuropathic pain [106]. Rapamycin can also be used as an autophagy inducer to slow down the chronification of neuropathic pain [104]. These studies suggest that the regulation of autophagy may be involved in the therapeutic effect of mTOR on neuropathic pain. Although there are few studies on autophagy of insular neurons, it may have potential value to explore the mechanism of neuropathic pain from this perspective. 

### 3.7. Oxytocin 

The insula is considered to be involved in pain empathy, which is the ability to perceive and understand the pain of others [32]. In recent years, pain empathy has attracted the attention of many researchers. The sociality of pain has also become an important issue. Oxytocin, as an important molecule in empathy research, has also started to gain favor with pain researchers. The microinjection of oxytocin into the RAIC of rats can reduce nociceptive behaviors induced by formalin injection of the hind paw and the spontaneous firing of spinal neurons [107].

In rats with partial ligation of the infraorbital nerve (pl-ION) causing neuropathic pain, a local injection of oxytocin into the injured infraorbital nerve combined with local low-level laser therapy inhibited the excitation of the somatosensory cortex and insular cortex induced by dental pulp electrical stimuli [108]. These studies confirmed the analgesic effect of oxytocin. Oxytocin has the potential to be used in the treatment of neuropathic pain. However, whether this analgesic effect is directly related to the insula requires further confirmation.

### 3.8. Other Agents

There are other molecular mechanisms involved in neuropathic pain processing in the insular lobe. A study observed increased concentrations of choline in the posterior insular cortex in rats with oxaliplatin-induced neuropathic pain [109]. As a centrally active acetylcholinesterase inhibitor, donepezil reversed the allodynia induced by oxaliplatin, and the microinjection of an M2 cholinergic antagonist into the posterior insular cortex attenuated the analgesic effect of donepezil [109]. Histamine regulates the sensory and emotional dimensions of neuropathic pain. In rats with the SNI model, the microinjection of histamine into the agranular insular cortex inhibited pain and aversion-related behaviors [110]. The injection of a selective inhibitor of protein kinase Mζ (which plays a key role in long-term potentiation) into the insular cortex reduced mechanical allodynia after peripheral nerve injury in rats [111]. These molecules could be important targets for insular analgesia.

## 4. Potential Sex-Based Differences

Neuropathic pain has demonstrated significant sex-based differences in many studies. The prevalence of neuropathic pain, such as TN, postherpetic neuralgia (PHN), complex regional pain syndrome (CRPS), and fibromyalgia, was shown to be higher in women compared with in men [112,113,114]. Coyle et al. used partial sciatic nerve ligation to establish neuropathic pain in rats and found that a larger proportion of female rats developed tactile allodynia [115]. In terms of analgesia, gender differences sometimes exist. For example, an antiepileptic drug called lacosamide had a stronger anti-allodynia effect on female rats with neuropathic pain [116]. Another study found that intrathecal neostigmine (a cholinergic agent) was more effective in reducing hyperalgesia in female rats with neuropathic pain caused by nerve injury than in male rats [117].

Some studies discussed the sex dimorphism of insula structure alterations in chronic pain, including irritable bowel syndrome (IBS) [118], migraine, and urological chronic pelvic pain (UCPPS). Although these types of chronic pain do not belong to neuropathic pain, they still have a certain reference value, because they have central sensitization symptoms, and their treatment methods are similar to those for neuropathic pain. A review summarized these studies and found that male patients had more obvious pain-related activation in the insular cortex compared to female patients [119]. In addition, female patients with migraines had greater cortical thickness in the posterior insula and more connectivity between the insula and certain brain regions [120]. Whether this insular gender dimorphism exists in patients with neuropathic pain remains to be explored.

The sex difference of insula was also explored at the molecular level. A study found that the dopamine (DA) and serotonin (5-HT) levels in certain brain areas (including the insula) in female rats were higher than those in male rats. The ratios of DA metabolite/DA and 5-HT metabolism/5-HT in the insula, as well as other regions of the male brain, were also found to be higher than for female rats [121]. This suggests that male rats have a higher metabolic rate of monoamine transmitters, which may affect gender-based differences in pain and other emotional processing.

Oxytocin (OXT) plays a more important role in sex differences. In a study of the social sharing paradigm, the activity of the insula decreased in women and increased in men after transnasal oxytocin [122]. This study also showed that OXT promoted the effect of women sharing positive experiences with others, but not for men [122]. In another study, intranasal oxytocin inhibited the activity of the male anterior insula induced by threatening social stimuli, but increased the activity in the female anterior insula [123]. These studies confirmed that oxytocin can affect the insula and that there are gender differences in this effect. The analgesic effect of oxytocin also has gender differences. A study proved that the intrathecal injection of oxytocin reversed allodynia in male mice but not in female mice after partial sciatic nerve ligation surgery [124]. Whether the analgesic effect of oxytocin also has an insular mechanism remains to be explored.

The effect of gonadal hormones on neuropathic pain has also been investigated. A study found that 17β-estradiol administration can partially rescue the poor prognosis of allodynia in female mice after CCI [125], which is a starting fact to develop gender-oriented therapy of neuropathic pain. In recent years, some researchers have focused on the therapeutic effect of estrogen on neuropathic pain and its mechanism [126,127]. However, whether the insula is involved in this process remains unclear. The relationship between the estrogen and insula has been explored. For example, the administration of estradiol on ovariectomized rats can influence the tail-hiding behavior and c-fos expression in the insula in the cold environment (16 °C) [128]. Whether the effect of gonadal hormone on insular activity also exists in neuropathic pain still needs further investigation. 

Numerous studies confirmed that opioid analgesics have gender differences. Rodent experiments showed that male animals are more sensitive to morphine analgesia than female animals [129,130]. A clinical study found that women had more intense pain and more morphine consumption after surgery than men [131]. However, studies of human subjects found inconsistent results. A study found greater morphine potency in female subjects, but this study focused on healthy subjects [132], which may be different from clinical and animal experiments.

Gonadal hormones may play an important role in the gender difference of opioid analgesia [133]. However, it is unclear whether there are gender differences in the effects of opioids on neuropathic pain. In addition, there is no direct relationship between the insula and the gender differences of opioid analgesia. This may be because opioids are more likely to affect the sensory dimensions of pain; however, for neuropathic pain, gender differences may be related to emotional, psychological, and social factors, and the insular lobe is more involved in these dimensions.

In recent years, social factors have been of concern in the field of pain [134]. Studies explored the relationship between pain and empathy [135]. The insula is also an important brain area for empathy. A study found that, when a healthy mouse lived together with a neuropathic pain mouse, it also produced nociceptive behavior responses, which could be reverted by inactivation of the insula induced by the bilateral microinjection of midazolam (a GABAA agonist) [32]. In future studies, the effects of empathy and sociological factors on gender differences in neuropathic pain and the role of the insular lobe could be further investigated.

## 5. Synesthesia and Potential Analgesic Effect of the Insula: Perspectives

As the center of information integration, the insula may play an important role in synesthesia processing. In 2006, a study reported odor and pain synesthesia in a male patient with neuropathic pain. Certain odors caused electric shock-like pain, which aggravated his neuropathic pain. Pain-related regions, including the insular cortex were activated in this process [28]. In 2010, Thomas-Anterion et al. reported a female patient with a lesion of the left posterior insula and SII. The lesion caused occasional and compulsive painting needs and typical neuropathic pain symptoms. Painting with cold colors increased the pain intensity [136]. The insular mechanism of synesthesia associated with neuropathic pain still needs to be further explored.

Electrical stimulation or lesions of the insula can help us to understand how the insular lobe participates in neuropathic pain. A study of posterior insular stimulation in cats found that 50 Hz insula stimulation increased the firing rate of non-nociceptive neurons and decreased the burst firing of nociceptive neurons in the thalamus [137]. Radiofrequency lesions of the RAIC decreased pain-related behaviors in rats with neuropathic pain caused by sciatic nerve ligation [138]. The insular cortex connects to a large number of brain structures and participates in the processing of sensation, emotion, cognition, and empathy. Direct electrical stimulation of the insula may have risks, such as causing other unpleasant sensations, emotions, and even improper behaviors. To solve these problems, a more precise understanding of insular functions may be needed.

From another perspective, this synesthesia phenomenon may suggest a possibility that other sensory inputs can also modulate pain, which may become the basis and exploration direction of alternative pain treatments. Exploring how the insula processes synesthesia between pain and other sensations may also help to explore this possibility. Mindfulness is a form of physical and mental mediation, emphasizing that the trainer pays attention to and maintains awareness of interoception, such as changes in breathing. As an important processing area of interoception, the effect of mindfulness on the insula may need to be investigated. Studies found that mindfulness meditation can relieve pain [139]. In the future, it should be considered whether the modulations of mindfulness on the insular lobe participate in analgesia. In addition, the role of the insula in other emotional and cognitive treatments of neuropathic pain may also be worth considering.

## 6. Conclusions

In this review, we described the structural and functional changes that occur in the insular cortex in the presence of neuropathic pain and after pain relief, as well as the related molecular mechanisms. We also discussed potential sex-based differences in response to neuropathic pain treatment on the basis of these molecular mechanisms. At present, the main problem is that the study of the insular lobe is still relatively limited, and there are many mysteries compared to the other brain regions. The multilevel and multidimensional information integration strategy of the insular cortex may be unique, and further research is needed to reveal its panorama.

## Figures and Tables

**Table 1 ijms-22-02648-t001:** Typical structural and functional changes of the insular cortex in neuropathic pain.

Changes	Subregion of IC	Types of Neuropathic Pain
Structure		
Decreased GMV	Anterior IC	Trigeminal neuropathic pain [34]
		Multiple types of neuropathic pain [35]
		Spinal cord injury [40]
	IC	Trigeminal neuralgia/trigeminal neuropathic pain [39]
		Spinal cord injury [41,42]
		Traumatic brain injury [44]
Increased GMV	Posterior IC	Trigeminal neuropathic pain [34]
		Multiple types of neuropathic pain [35]
Cortical thinning	Dorsal posterior and ventral anterior IC	Trigeminal neuralgia [36]
	IC	Ankylosing spondylitis back pain [43]
Function		
Increased activity	Caudal anterior IC	Fibromyalgia [48]
	Anterior IC	Sciatic spared nerve injury [50]
		Brush-evoked allodynia in multiple types of neuropathic pain [53]
		Thermal hyperalgesia in diabetic neuropathic pain [54]
	Medial IC	Cold allodynia after subcutaneous injection of ciguatoxins [51]
	IC	Brush-evoked allodynia in peripheral nerve injury [52]

GMV, gray matter volume; IC, insular cortex.

## Data Availability

Not applicable.

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
