# Peer review of "Current Understanding of the Involvement of the Insular Cortex in Neuropathic Pain: A Narrative Review"

_ijms, 2021, doi:10.3390/ijms22052648_

Round 1
Reviewer 1 Report
In the paper entitled:”Current understanding of the involvement of the insular cortex in neuropathic pain: a narrative review” the authors describe structural and functional changes, related molecular mechanisms and potential sex-based differences that occur in the insular cortex in the presence of neuropathic pain and after pain relief. The work makes a honest contribution to the field but, in my opinion it needs a major revision.
First of all it requires an accurate revision of the English language in order to avoid errors of the type “are increases pag 1 line 29” and, above all, improve the fluidity of exposure (for example page 2 line 59: “one study described a case that there are some confluent small foci of softening in the insular cortex and parietal tectum of a patient with neuropathic pain symptoms”)

Page 1, line 34: Given the importance of data relating to the use of opioids in neuropathic pain, it would be appropriate at this point to enrich the related bibliographic data not limiting itself to saying that opioids have limited effects but by introducing other bibliographic data regarding selective agonists of the opioid receptor subtypes such as “Repeated activation of delta opioid receptors counteracts nerve injury-induced TNF-α up-regulation in the sciatic nerve of rats with neuropathic pain: A possible correlation with delta opioid receptors-mediated antiallodinic effect Vicario N., et al. (2016) Molecular Pain, 12 , pp. 1-7”.
The same for page 2 line 80: I agree with the authors' position not to go into the specifics of the mechanisms of the other areas involved in neuropathic pain but it seems to me necessary to include some bibliographic data at this level.
Page 3 line 10: the authors speak of functional alterations indicating them as activation, but what does this activation consist of? how is it measured? (Except that the word activation is repeated countless times in all the text…)
Page 4 line 137: “decreased inhibitory neurotransmitter content “
but to which neurotransmitters do the authors refer?
Page 4 line 145: “These regions may belong to the salience network” 
What is this network?
Page 6 line 170: What do the authors mean by this title? “Molecular Expression in the Insular Cortex in the Presence of Neuropathic Pain”? line 172: for “the expression of analgesic-related molecules in the insula” the authors mean different types of receptors? Express more clearly. Then in 3.1 (line 177) replace “opioids” with opioid receptors and so for the other paragraphs.
Page 5, line 204-210: The authors list the effects of a series of substances and not of cannabinoid receptor. Only in the conclusion they cite a possible increased expression of the cannabinoid receptor in the insula, documented, as indicated in the bibliography, from another bibliographic data.
Page 6 Line 212: “the insular administration of the CB1 receptor”???

Page 6 Line 213: “The interference of neuropathic pain with the CB1 receptor”? Do you mean 
” the involvement of CB1 receptor in the modulation of neuropathic pain”?
Page 7 line 304-310: Are the examples cited related to insula?
Page 8 line 324-335: If the bibliographic notes are not related to neuropathic pain why are they cited?
Line 349: It would be better to insert the bibliographic data where the author or year is inserted
The work also requires careful rereading to eliminate numerous careless errors
Reviewer 2 Report
The author present a narrative literature review about the involvement of insula in the neuropatic pain. The authors also discussed some potential gender differences underlying the onset of neuropatic pain. Whereas the topic is undoubtely intriguing, I think that this study present relevant issues that greatly affect the present paper. The main limitation is that insula appears a "lonely island" inside the brain; it is really difficult to understand its role outside a more wide brain network: the authors introduce this concept in the section 2.3 "Brain network involvement", but this is the pivotal point to analyze the insular role in pain modulation. The authors report that thikness of insular region at neuroimaging could justify an primary (and lonely) involvement of the insula in pain modulator networks, but this should be considered in the contexct of more wide parenchyma structural alteration. In other words, there are some experiences of neuropatic pain in which the only alteration is charged to the insular lobe alone?
Also, the gender analysis present many issues: "There are also sex differences in the analgesic effect of opioids on neuropathic pain. But the difference is not consistent [94]. A prospective cohort study found that females had larger morphine consumption after surgery than men [90]. But there are also opposite results [95]." Therefore, it is hard to understand the scientific value of this consideration. Moreover, this gender analysis is based on a couple of experimental murine model, and on two neuroimaging studies, that contradict each other.
lines 36-37: "Surgical treatment and deep brain stimulation also may be options for pain control". The invasive treatment for neuropatic pain, despite is limited option, comprises for example motor cortex neuromodulation, or DREZ. If the author would introduce this concept, they should clarify how the neuromodulation of fronto-parietal area could impact the insular lobe activity.
Round 2
Reviewer 1 Report
The paper has been improved in its various parts. I have no other comments on thisAuthor Response
Thanks a lot for the Reviewer’s comment.
Reviewer 2 Report
The article has greatly revisited, according to reviewers suggestions. I think that the scientific value had increased, whereas an English editing still remains necessary, mainly for syntax
Author Response
Thank you very much for your advice. In order to solve the language problem, we checked the manuscript thoroughly and invited an English language editor to help us revise it. All changes in the revised manuscript are highlighted using the "Track Changes" function in Microsoft Word.
Round 3
Reviewer 2 Report
The authors have greatly revised the manuscript, and I think that it is now suitable for publication
Author Response
Thank you very much for the reviewer's comment.